# Milwaukee Makes a Difference: Recognizing Gifted Students from Culturally and Linguistically Diverse Families

Susan O'Brien, Mary Ruth Coleman *, Dorothy L. Schuller, Martha A. López and German Díaz

Department of Curriculum and Instruction, Milwaukee Public Schools, 5225 W. Vliet Street, Milwaukee, WI 53208, USA; obriensj@milwaukee.k12.wi.us (S.O.); schulldl@milwaukee.k12.wi.us (D.L.S.); lopezm2@milwaukee.k12.wi.us (M.A.L.); diazga@milwaukee.k12.wi.us (G.D.)
* Correspondence: maryruth.coleman@unc.edu

**Abstract:** Gifted education today faces a significant challenge in reaching equity as well as excellence. This is reflected in the disproportionate underrepresentation of children from Black, Hispanic, Native, and low-income families. This pattern of underrepresentation within programs for students with gifts and talents is pervasive and pernicious and impacts gifted education programming across all 50 states in the United States of America. This article describes the efforts of Milwaukee Public Schools, a large urban school district in Wisconsin, to address the need for both equity and excellence within their gifted education programming. The U~STRARS~PLUS model formed the foundation for changing the culture of the schools from "at risk" to "at potential". Dedicated leadership and the combination of securing external support, developing internal trust, and building capacity across the district were critical to creating a strength-based focus within the schools. While the journey is not over, the authors hope that others can learn from Milwaukee's experiences.

**Keywords:** gifted education; cultural diversity; linguistic diversity; U-STARS; equity and excellence





## 1. Introduction

One of the greatest challenges facing gifted education in the United States today is the disproportionate underrepresentation of children from Black, Hispanic, Native, and low-income families. This pattern of underrepresentation within programs for students with gifts and talents is pervasive and pernicious, and the board of directors of the National Association for Gifted Children has acknowledged the role of structural and systemic racism as contributing factors to the inequities in identification of students with gifts and talents:

"We acknowledge the injustices of structural and systemic racism and recognize the field of gifted education has historically been part of the problem by promoting these injustices, even if inadvertently. Some early researchers and thought leaders who influenced the field were involved with the eugenics movement, and early gifted identification and programming practices often became vehicles for de facto segregation. The field has made tremendous strides in addressing these historical injustices in recent years, but we have not made sufficient progress". (National Association for Gifted Children, 2020, para. 3) [1].

Students from Black, Hispanic, Native, and/or low-income families are significantly underrepresented within gifted education programs (Rinn et al., 2020) [2]. Conversely, Asian and White students are overrepresented proportionately within gifted education programs (Hodges et al., 2018) [3], showing the disparate impact that identification policies and procedures have on students from different racial and ethnic groups. Determining the extent of the problem of disproportionality is difficult because gifted education is not covered by federal law, so gathering and reporting data on who is identified as gifted varies widely from state to state (Rinn et al., 2020) [2].

In a large study of identification practices, Hodges et al. (2018) [3] identified the "risk ratio" for Black, Hispanic, and Native American students being identified for gifted

education as 0.34. In other words, if you are a Black, Hispanic, or Native American student, you are about one-third as likely to be identified for gifted education as if you are an Asian or White student. Disproportionate underrepresentation within gifted education services is seen across all states (Rinn et al., 2020) [2].

The challenge of disproportionality is exacerbated because children of color and poverty often attend schools with the fewest resources, and these schools may embody the pedagogy of poverty—for example, a deficit approach to learning with didactic teacher direction; limited student choice, enrichment, or exploration; and a focus on remediation (Coleman, 2016) [4]. High-poverty schools may also find it difficult to establish strong partnerships with families, and this further undermines the child's success.

The challenge of disproportionate representation that we face in gifted education, however, goes well beyond just identification decisions (Ford et al., 2018) [5]. Our challenge is how we can shift the education system toward a strength-based approach that supports the expansion of excellence to all students, ensuring that each student receives an appropriately stimulating and enriched education. This demands the transformation of gifted programs and the way that schools assess students' gifted potential, changing a system that, in essence, has been exclusionary of minority students to one that is built on strength-based, culturally relevant teaching practices. In this article, we share Milwaukee's journey in response to this challenge.

## 2. Overview of Milwaukee Public Schools

Milwaukee Public Schools (MPS) serves the urban area of Milwaukee, Wisconsin, which is home to approximately 600,000 people. In 112 of MPS schools, there are approximately 15,525 students in grades 1–3, so making changes to improve services for students with gifts and talents seemed daunting. Most recent data from the Civil Rights Data Collection (CRDC, 2017) [6] for MPS show the demographics of the total enrollment ($n = 75,753$). The largest ethnic group represented in MPS are Blacks with 52.1% of the enrollment, followed by 26.6% of Hispanics, 10.8% Whites, 7.1% Asians, 0.6% American Indians or Alaska Natives, and 0.1% Native Hawaiians or Pacific Islanders. Overall, approximately 90.1% of students in MPS are students of color, 84.1% are economically disadvantaged, and 20.2% are students with special needs (Milwaukee Public Schools, 2021) [7]. Despite the pattern of "minority–majority" in student diversity, MPS follows national trends when looking at the issue of underrepresentation of Black, Hispanic, and Native students within gifted and talented programs.

Currently, MPS uses the Cognitive Abilities Test (CogAT) as a universal screener in second grade to determine whether students are gifted and the Teacher's Observation of Potential in Students (TOPS) as a tool to identify gifted potential.

Using the CogAT data alone for the school year 2019–2020, gifted and talented students would make up only 0.61% ($n = 440$) of the total MPS student population. When these data are disaggregated by ethnicity, the results show that for CogAT-identified gifted students, 54.3% are White, 20.9% are Hispanic, 10.9% are Black or African American, 10.0% are Asian, 3.2% are multiple races, and 0.7% are American Indian or Alaska Native. The underrepresentation of students with limited English proficiency and with disability status is concerning as well. The challenges that MPS faces reflect those of many districts, and, in sharing our story, we hope to help others as they work to address inequities across gifted and talented programming.

## 3. U-STARS~PLUS: A Strength-Based Approach to Nurture, Recognize, and Respond to Giftedness

U-STARS~PLUS (Using Science, Talents, and Abilities to Recognize Students ~ Promoting Learning for Under-Represented Students) is a strength-based approach focused on bringing together equity and excellence (Coleman, 2016) [4]. The five components of this approach are creating a "high-end" learning environment within the general classroom, systematically observing students to recognize their strengths, using hands-on science activities as a platform for observations, engaging families in meaningful partnerships



to support students' success, and developing the infrastructure needed for successful implementation by expanding the capacity of building and district leadership, attending to fidelity of implementation, and using data to review accomplishments (for a full description of the U-STARS~PLUS approach, see Coleman, 2016 [4]).

A key element of U-STARS~PLUS is TOPS (Coleman et al., 2010) [8], used to help teachers see students' strengths. The TOPS is an instrument which supports teachers in their systematic observations of student's strengths across nine domains: learns easily, shows advanced skills, displays curiosity and creativity, has strong interests, shows advanced reasoning, displays spatial abilities, shows motivation, shows social perceptiveness, and displays leadership. Strengths in these nine domains are an indication of high-potential which should be nurtured. Observing for strengths, particularly with student populations who are underserved within gifted education, involves a shift in perspective from "at risk" to "at potential". With an "at risk" view, interventions focus on minimizing risk and remediating deficits, while the "at potential" view focuses on maximizing potential and creating environments that respond to strengths (Coleman, 2005) [9]. This shift in perspective is profound. The famed astronomer Johannes Kepler (1571–1630) is reported to have said, "Be careful how you perceive the world . . . it is that way". When we see our students as capable, interesting, and smart, we design our learning environments to support children who are capable, interesting, and smart!

The strength-based approach of U-STARS~PLUS hinges on an iterative process of nurturing potential in all our students, recognizing our students' strengths, and responding to these strengths with additional scaffolded learning experiences to provide progressive challenge and support for growth. One intended outcome of this approach is to increase the representation of students from Black, Hispanic, Native, and low-income families in gifted and advanced-learner programs. The broader aspiration, however, is to change the culture of our schools. This shift in school culture can be seen in the attitudes and practices of educators as we move toward an "at potential" mindset with our students, their families, and each other.

## 4. Milwaukee's Work toward Equitable Gifted Education Programming

Systemic change is hard work, and it takes time. Our journey in MPS started in 2013 with a small grant from the Wisconsin Department of Public Instruction (DPI). At that time, the primary focus of "gifted education" in MPS was to oversee the Advanced Placement and International Baccalaureate programs in high school and to offer enrichment resources on an "as needed" basis when schools, teachers, or parents contacted the department with a request. The grant from the DPI allowed MPS to begin rethinking services for younger students with gifts and talents. The first step was establishing a screening process for identifying students in the second grade who would benefit from gifted education support. Right from the beginning, the goal was to develop an equitable identification process that would allow for the recognition of students with high potential from across the diverse student body of MPS. The CogAT, in both English and Spanish, and TOPS from U-STARS~PLUS formed the core of the screening and identification process.

Pilots of the U-STARS~PLUS model were begun in two elementary schools: Allen-Field and Victory. At both schools, the grades 1–3 bilingual and monolingual teachers were involved in an abundance of professional development in the U-STARS~PLUS framework that included high-end learning opportunities, teacher's systematic observations, hands-on/inquiry-based science, family and school partnerships, and infrastructure for building systemic change (Coleman, 2016) [4]. The positive impact was apparent almost immediately. The teachers embraced the framework, and it was evident that their mindsets began to shift from an "at risk" lens to an "at potential" lens. This new perspective led to providing rich learning opportunities for students using inquiry-based learning, particularly by merging science and literacy. As a result, district assessments demonstrated a significant closure of the achievement and excellence gap while at the same time increasing the number of students identified as having gifted potential using TOPS. Recognizing young student's

strengths and nurturing their potential may lead to formal identification as "gifted and talented" as they progress through school (the identification process is described later in the article).

Parental/family engagement was the next focus, and this entailed building a communication network with the parents of students identified as gifted and talented as well as students with high potential. This network was critical to share information about "giftedness" and to let parents know what opportunities were available to their child both within MPS and across the region. Involving parents at this early stage in the program's development was seen as key to building support for the students' success. In 2014, MPS held its first family event. This event provided information for parents, including the MPS process for screening through CogAT, characteristics of gifted children, and resources available to them. Students were gathered in a separate room to experience activities based on high interests and creativity.

By 2015, MPS held its first summer enrichment opportunity—"Camp Invention"—for students in grades 1–5 who identified as gifted and talented. This work was led by Dorothy Schuller, an Advanced Academic Programs coach. Creating an engaging and enriching camp experience for our students involved program design using best practices for gifted learners, teacher professional development, and careful and thoughtful selection of materials. With a screening process, a parent network, and the summer camp in place, we felt we had made progress—but knew that there was far more to be done.

## 5. Expanding Excellence Grant 2015–2018

In 2015, the DPI was awarded a $1.1 million, three-year Jacob K. Javits Gifted and Talented Students Education grant from the U.S. Department of Education. The Javits Expanding Excellence project was in response to the state and national need to reduce the disproportionality in identification of students for advanced learning services. The purpose of the grant was to find and nurture students with high potential in grades 1–3 using a strength-based approach; the specific focus was to mitigate disproportionality in students identified for gifted services and to reduce the excellence gap. Milwaukee Public Schools was one of three districts included in the project. The grant incorporated the U-STARS~PLUS components, including TOPS, the science and literature model lessons, and the family engagement science packets. Central to the grant was the preparation of MPS teachers and administrators to implement culturally and linguistically responsive, strength-based practices.

Specifically, this project intended to expand the expertise of demonstration site staff regarding best practices in gifted education with a targeted emphasis on how best to meet the advanced learning needs of two groups of disadvantaged students: those who qualify for free or reduced-price meals and English language learners. The goals of the project focused on the areas of collaboration, assessment, and instruction.

Milwaukee Public Schools identified ten demonstration schools for this phase of implementation. Demonstration classroom teachers received professional development in, among other things, analyzing data for disproportionality, implementing a Response to Intervention (RTI, also called Multi-Tiered System of Supports) framework that includes services for high-ability/high-potential students, culturally responsive practices, and the U-STARS~PLUS resources. Demonstration classroom teachers were offered the opportunity to participate in a state-approved educator licensure program in gifted education. A leadership cadre was formed at the state and district levels to guide the work of the grant.

With the demonstration schools in place, the MPS leadership cadre began the process of professional development for participating teachers of grades 1–3. Demonstration school participating teachers as well as other leadership cadre members were given the opportunity to enroll in UW–Whitewater gifted and talented teacher/coordinator licensure courses at a deferred cost. This intense coursework allowed key grant participants to learn and grow while engaging within the grant; at the end of the two-year program, twenty-five staff members were granted licenses.

The needs of advanced learners were incorporated into the schools' RTI planning, and documentation strategies were put in place for the school improvement plan. Incorporating advanced learners within the RTI process and school improvement plans raised the commitment to and accountability for addressing the needs of these students.

Parents and guardians of identified children were invited to a mid-year informational session at MPS Central Services. The parent event allowed staff from Advanced Academic Programs to explain the process of the CogAT and TOPS and give an in-depth presentation of the data that each assessment showed. Resources were shared for parents to utilize enrichment pieces at home as well as show what each of their schools has to offer. Gifted camps were offered free of charge in the fall and the spring on Saturday mornings and during two-week sessions in the summer. These camps were funded through the grant, which covered camp materials and teacher pay. Parent workshops on characteristics of gifted children were offered in the fall and spring as well. These sessions were four to six weeks in duration and included topics on motivation, twice exceptionalities, student advocacy, perfectionism, and depression.

With all the work being done in collaboration with the Expanding Excellence grant, there was still more to accomplish and, continuing to build on successful practices, district leadership committed to writing our own Javits grant application: the Scaling-up and Expanding Excellence for Underrepresented Students (SEE US!) grant in 2017. With SEE US! MPS took the work to a new level and the practices became established in the demonstration classrooms, thus reaching even more students and empowering more teachers. This is mainly due to the insightfulness of the grant application authors who created a fulltime position of a grant coordinator to design, implement, and oversee all aspects of the grant. When discussing the uniqueness of this position, Susan O'Brien, the MPS Javits grant coordinator shared,

> From what I understand from networking with others in the GT, grant, and Javits communities, it is unusual to have a fulltime position dedicated solely to this work. This structure is why we have been able to launch and ultimately institutionalize the goals of the SEE US! Grant and build lasting relationships with teachers, school leaders, and families. In my position, the work is prioritized every day. It's what our students deserve and I am honored serve as the grant coordinator.

## 6. Scaling-Up and Expanding Excellence for Underrepresented Students (SEE US!) Javits Grant Award 2017–2022 ($2,261,685)

The SEE US! grant, like its predecessor, focused on closing the excellence gap for economically disadvantaged students—primarily students of color—using an RTI framework and the evidence-based strategies within the U-STARS~PLUS hands-on, inquiry-based science and literacy units, lessons, and high-end learning opportunities. The goals of the program are to increase the level and depth of collaboration among school and district personnel, students, and students' families to support the academic success of students from economically disadvantaged and culturally different families; to increase the number of high-ability/high-potential students from economically disadvantaged and culturally different families identified for advanced services; to increase the percentages of students who achieve at advanced levels in reading and mathematics; and to continue full implementation for at least three years after the grant.

SEE US! funding was used to support professional development, adding 30 more participating classroom teachers (grades 1–3), who serve approximately 900 students, to deepen and extend best practices across the district. At the time of this article, there are more than 100 teachers and administrators participating in grant implementation and more than 1300 students benefiting from implementation. This professional development built on and augmented the earlier workshops. Appendix A shows the combined professional development offerings of the SEE US! and SURGE Javits grants. The coordination of professional development, sustained across multiple years, has allowed MPS to develop

a more knowledgeable workforce with the skills and abilities to nurture, recognize, and respond to the needs of students with high potential within the general education classes. Professional development to build capacity has been central to supporting positive change across the implementing schools.

Each professional development session had built on the previous session and incorporated participant feedback, interests, and requests for additional information. By the end of this first round of professional development sessions in 2018–2019, participant educators had developed a solid grounding in identification, curriculum modifications, and instructional methods and programming for underrepresented gifted students. This set the stage for more specialization in later years of the project.

With the ongoing success of the two Javits grants—Expanding Excellence and SEE US!—Susan O'Brien, MPS Javits grant coordinator/project manager, and Dorothy Schuller, a gifted and talented-certified colleague, authored and applied for another Javits grant for MPS with the goal to nurture, recognize, and respond to even more underrepresented students identified with high ability/high potential.

## 7. Serving the Underrepresented by Grouping Equitably (SURGE) Javits Grant Award 2019 ($2,929,319)

The school selection criteria for SURGE supported a larger district-wide plan to ensure that the demographics of the district were represented in the demographics of gifted and talented program opportunities across the city. Each SURGE school met the following selection criteria for participation: (1) the school is in a geographic location of the city where traditionally underrepresented families reside, (2) the school leader and staff are committed to supporting the gifts and talents of their students, (3) the school's student demographic groups are not proportionately reflected in the district's gifted and talented programs, (4) the school maintains an economically disadvantaged rate greater than 60%, and (5) the school has minimal access to high-quality specialty gifted and talented programs. Eight schools were recruited to participate in this newly awarded Javits grant. With the advent of COVID-19 restrictions, the work of SURGE had to pivot to online support. The SURGE participants joined with SEE US! participants in professional development the fall of 2020 (see Appendix A). SURGE is now under way with initial professional development and steps toward implementation of TOPS. During the summer of 2020, the implementation team began preparing for possible virtual school in the fall of 2020. Time was spent developing virtual identification tools and preparing teachers to create virtual observable environments to successfully identify gifted potential in their students. This forward-thinking approach led to successful grant implementation.

## 8. Building on Success: Review of the Impact on Milwaukee Public Schools

The initial seed money, given by the DPI in 2013, placed MPS on a journey that has spanned the last eight years. Starting from a base of few to no services for young students with high potential, the initial work at two elementary schools showed that early nurturing and recognition of potential was not only possible—it was practical. This led to MPS inclusion in the DPI Expanding Excellence grant and the development of a district leadership cadre to support the work of ten demonstration schools. Building on this success, the SEE US! grant expanded the professional development opportunities to create a knowledgeable and skilled implementation team. Susan O'Brien, MPS Javits grant coordinator, states,

> The grant implementation and professional development were carefully planned and strategically implemented for success. Nothing was "one and done". Following the professional development experiences, the team provided hands-on support while empowering teachers as professionals. I recall many conversations with teachers who told me how grateful they were for the professional development and how they welcomed the joyful rigor of the SEE US! grant practices.

As more teachers and administrators came on board, their success strengthened the foundation for lasting systemic change. The SURGE grant was built on this strong foundation and is positioned to extend the work even further.

This work did not progress serendipitously; it took dedicated and intentional effort to create the team needed to expand the work to thirteen more schools. Building trusting relationships has been a cornerstone of all the Javits grant work. It is important for the classroom teachers to know that the implementation team is composed of teachers and that the team is there to support them, not to evaluate their practices. Susan O'Brien, MPS Javits grant coordinator, says,

> Teachers know that we are there for them and that we are always willing to co-construct inquiry lessons, co-teach, and help support student engagement. We work hard to get them everything they need to ensure engaging, enriching, hands-on lessons.

With this trust in place, teachers felt free to engage in an authentic discourse as they worked to improve their practice.

Teachers and schools received a variety of supports in addition to formal professional development. After learning about the U-STARS~PLUS lessons and their alignment to the existing district instruction expectations, teachers were given budgets to purchase materials for hands-on science lessons (U-STARS~PLUS lessons) and inquiry-based instruction. Monthly meetings were set up by the MPS Javits grant coordinator at each school site with demonstration teachers, school support teachers, and the school administrator. These meetings evolved into a critical practice in successful implementation of the SEE US! grant. Collaborative discussions focusing on professional development follow-up, student identification practices, and inquiry-based instruction were the central themes of these meetings. According to Susan O'Brien,

> These monthly meetings were a game changer because the teachers, administrators, and I had the chance to build positive relationships, discuss grant implementation, and have insightful conversations about students with gifted potential and how to nurture that potential.

This level of trust and support resulted in observable changes in classroom practices, which aligned with the grant expectations. Formal evaluations of the SEE US! and SURGE grants are under way and will be reported elsewhere. Our goal here is to share the ongoing and dynamic observations of change that have been made by participants and external observers. Below are the perspectives of a project principal and three teachers. All have been a part of this work for all four years of the SEE US! grant.

## 9. One Principal's Perspective

Principals play a key role in the successful implementation of the Javits grant. Christlyn Frederick-Stanley, principal of Keefe Avenue School, shared the impact that the SEE US! grant has had on her professional growth and her school community. In 2017, her journey with the SEE US! Javits grant began when she received an invitation from the superintendent. She recalls being shocked to have been invited to such a great opportunity because of the bad reputation that schools have in her school's zip code.

The success of MPS has been possible because of principals like Frederick-Stanley and their commitment to excellence. This was manifested in Frederick-Stanley's personal desire to transform her predominately African American school into a caring and nurturing environment where all students would succeed. She believes in the power that schools have to transform the way that students are perceived, moving from being deficient to being strength-based. To exemplify this, she says, "Our job [as principals] is laying the foundation. We need them [students] to feel confident in their abilities to learn—period". Frederick-Stanley saw potential in the students at her school, yet she argues that lack of access to resources determines students' success, especially when potentially gifted minority students come from economically disadvantaged families. She acknowledges

that her "role [as an administrator] is more serving as an encourager". She sees her role in the grant as being a role model by participating in professional development and being actively involved in what teachers need to learn. She states that, as an administrator, "If I don't believe in it, I cannot speak the language, and therefore I cannot encourage others to do it".

Further, she adds that transforming a school culture and asking teachers to change teaching practices is not easy. When referring to inquiry-based learning, she understands that "for some teachers . . . they see it as an add-on, but for others, they say, 'I realize there are things I'm going to have to do differently in my classroom to promote this.'" In other words, she emphasizes that there must be a transformation, and teachers cannot keep doing the same things and expecting different results.

Although TOPS is a tool that serves to identify gifted potential in students, it also serves to challenge a teacher's traditional view of students. Frederick-Stanley acknowledges that issues of bias may prevent teachers from recognizing true potential in minority students. According to her, teachers who use the TOPS tool have demonstrated a change in mindset. Furthermore, she notes that "Our most successful teachers are the ones that embrace this as part of our school culture".

Referring to the SEE US! Javits grant training, she says, "For me, it reaffirmed the need for differentiation and having a greater appreciation for the fact that our students do learn differently, and if we are going to help them achieve at a higher level . . . we have to validate their style of learning rather than constantly making them conform to a different way".

Finally, Frederick-Stanley believes that the TOPS tool "really provokes and promotes the type of teaching and environment in a classroom that's far healthier than what we currently have. It opened doors and allowed light to shine in places where it wasn't shining".

## 10. Three Teachers' Perspectives

What has contributed to make MPS Javits schools a success? One may assert that what makes schools successful is the work that teachers do to nurture and develop students' gifts and talents. In this section, Michelle Taylor, Martha López, and Germán Díaz shared with us their experiences and the impact that the Javits grants have had on them.

First, Taylor works at Congress School, a K–8 SEE US! school whose mission is to "create a well-prepared and caring community of learners in which students work hard to be successful and learning never stops". Taylor is a veteran teacher of twenty-one years and joined the SEE US! Javits grant in 2018. She describes her participation in this grant as a transformative and empowering experience. She explains that "The Javits SEE US! grant has blossomed and lifted me [and] allowed me to be who I know I can be [as an educator]". Her participation in the SEE US! grant has contributed to a change in her mindset to become more inclusive about who could be gifted. She notes that often gifted students with non-traditional teaching pleasing behavior are being overlooked, which contributes to the issue of underrepresentation. She says,

> I no longer look at students through their negative behaviors. I look at students with a more positive light. The biggest impact [on me] is that I look at students through a totally different lens. I often wonder how I can use my students' strengths into something positive.

When reflecting on her journey as an educator, she is very grateful for the opportunity she has had to participate in professional development where she learned to use the TOPS tool to identify, nurture, and develop gifted potential both in students with pleasing and non-pleasing behaviors. Taylor advises that "Every teacher needs to be taught the TOPS tool. Our students are creative and amazing and have their talents, and I believe every person has a hidden talent". Her conviction that every student can succeed has also challenged her to learn new skills, such as coding and becoming a LEGO® robotics coach, which has resulted in an increase of access to learning opportunities for her students. For

example, she recently created a class on coding for the purpose of "teaching new skills and engaging students who might be introverts".

Teacher number two, López, a former bilingual teacher and currently a SURGE grant teacher coach for MPS, supports current K–grade 3 SURGE Javits grant teachers. Her passion for advocating for underrepresented gifted and talented students stems from her participation in the Expanding Excellence Javits grant in 2015 at Allen-Field Elementary School. She was a second-grade bilingual teacher and was excited to learn about the U-STARS~PLUS framework. She reflects that

> Being part of the Expanding Excellence Javits grant impacted me professionally and personally. As I began integrating literacy and science into my daily teaching practices, I realized how enthusiastic I was about teaching. Teaching was no longer a list of standards I had to check off—rather, it was an opportunity to engage in real-life learning alongside my students.

These rich learning experiences were positively impacting her students too. For instance, she discussed that students' attendance increased significantly, the majority of her students were scoring "proficient" and "advanced" in class and on district standardized assessments, and students made a psychological investment in their learning. As a result, her personal perception of what a gifted and talented student looks like, sounds like, and talks like suddenly crumbled and reconstructed to include Latino students who were underprivileged, non-English speakers, or non-pleasing students. For López, gifted and talented students demonstrated their abilities far beyond taking a test. She says, "Gifted students are leaders who stand up for what they believe. They are creative thinkers, problem solvers, and innovators". Furthermore, she stresses that bilingual gifted students skillfully navigate two cultures and two languages, and they impressively adapt to their environment. To her, this subpopulation of gifted learners is often overlooked and demands a response from districts, researchers, and practitioners.

Díaz, teacher number three, works as a SURGE grant teacher coach for MPS. He realizes that the benefits of participating in these grant initiatives are many. He says,

> I have grown as an individual in the sense that I like to think of myself as a role model or someone who continues to mentor others about the huge potential that our students have. My commitment to advocate for the inclusion of minority students in traditionally white, middle-class gifted programs has resulted in researching and learning more about the richness and the diversity of talents that our students have.

He mentions that his participation in the Expanding Excellence Javits grant while in the classroom had huge effects on his students by increasing access to opportunities and provisioning multiple venues to nurture talent development beyond academic skills. He says,

> It [the grant experience] impacted me greatly. But I think it impacted my students the most. This has occurred in many different ways, such as advocating to attain more resources and access opportunities to learn—Saturday camps—or simply by providing teachers like me with professional training that resulted in becoming a better teacher.

He adds that a huge change in mindset from seeing students as being "at risk" to being "at potential" has allowed him to

> . . . learn more about the diversity of talents and potential that many of my students already bring from home.

In sum, the experience of being part of the Javits grants on Taylor, López, and Díaz was impactful both professionally and personally and played a critical role in their development as educators and leaders. In 2019, MPS was awarded a new SURGE Javits grant. When asked what advice Taylor would give to new Javits grant teachers, she said,

Do it! The [number] of resources you get, whatever you can dream of, you get. If you're a teacher that has a spark . . . it's an amazing opportunity if you want a better understanding of how to identify your children and what to look for . . . it makes you a better teacher to better serve our students!

## 11. External Observers

As part of the overall evaluation, our external evaluator, Wisconsin Evaluation Collaborative (2021) [10], observed classroom practices to see what impact the professional development was having. In this observation, they were looking for evidence of the use of Bloom's Taxonomy:

In most of the lessons, the first four elements of Bloom's Taxonomy were present—Knowledge/Remember, Comprehension/Understand, Application/Apply, and Analysis/Analyze. These elements reflected the general progression of the lessons: Remember what you have learned before, understand what you are learning now or make predictions, apply what you have learned, and analyze why it did or did not work or what you think will happen. In the chrysalis activity, Synthesis/Create and Evaluation/Evaluate were also evident as the experiment progressed to completion.

Evidence of professional development was also observable in the demonstration classrooms through differentiation practices. The project's external evaluator observed the following:

Teachers differentiated their instruction in several ways, mostly through their purposeful grouping of students into small teams. The teachers who did the U-STARS lesson on animal habitats formed groups based on behavior with a group leader. Others used already-existing groups, such as those formed for math or reading. Several of the activities included learning centers or stations. Questioning techniques promoted higher-level thinking by having students make predictions about what may happen, in one case by utilizing turn-and-talks.

Furthermore, teachers created opportunities for students to engage in hands-on/inquiry-based work aligned with the U-STARS~PLUS lessons. The professional development and materials supplied by the Javits grant supported these instructional practices:

Students had ample opportunities to do hands-on, inquiry-based work across all the lessons described above, as they all involved experiments or observations and various materials that lent themselves to hands-on learning. Teachers asked students to hypothesize and encouraged them to ask questions. The units involved multiple subjects, often requiring students to read a text (or read along with a teacher) before doing the interactive science-related activities. Students also had the chance to develop new knowledge, skills, and vocabulary around animals and plants.

Evidence of cultural relevance or responsiveness was more difficult to document as it is harder to directly observe in a short lesson. In the unit on how animals eat, however, the teacher told the students that it is sometimes hard to find wildlife in their neighborhoods. The teacher showed a picture of a city street and told students that they can see birds, even if they cannot see chameleons or fish. The Wisconsin Evaluation Collaborative observer also noted a few examples of where students appeared to be interpreting lessons through their own contexts and lenses—growing up in a city. For instance, in the habitat lesson, a student referred to a burrow as being under "concrete" rather than earth.

## 12. Evolution of Practices toward a Strength-Based Approach

The changes that have taken place as a result of the focus on strength-based approaches to nurture, recognize, and respond to the needs of MPS's students with high potential have impacted the district, the schools, the teachers, the students, and their families.

These changes are reflected in the policies and practices that have been adopted by the participating grant schools.

## 13. District Impact

As teacher mindsets shifted toward an "at potential" view of their students, we began to see more students nominated for enrichment opportunities. Seeing students as "at potential" is the critical first step toward changing instruction practices to address students' strengths (Coleman, 2005) [9]. Figure 1 shows the increase in teacher nominations, using TOPS, of students with high potential across the years of implementation.

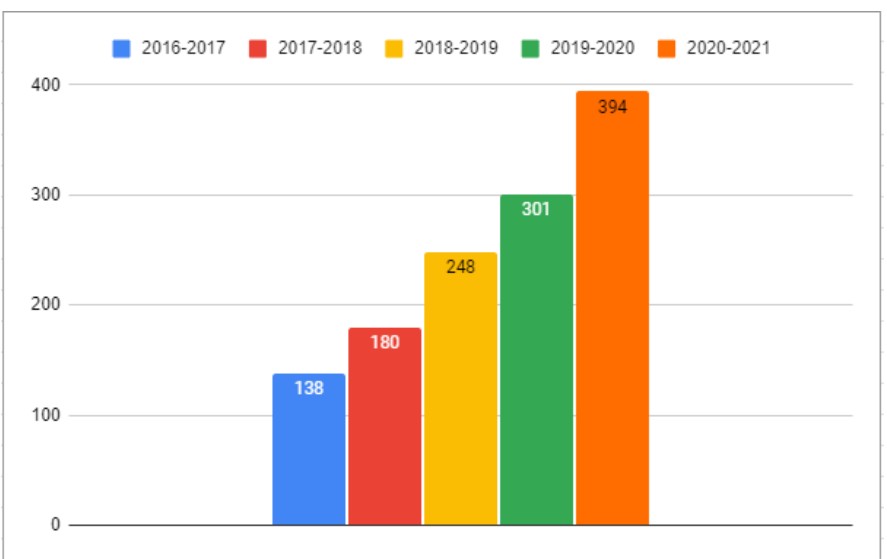

**Figure 1.** Increases in TOPS Nominations across the Years.

Seeing the increasing numbers of students who are seen by their teachers as having gifted potential is a positive trend, but the question remains as to how this trend is impacting students who have historically been viewed through a deficit "at risk" lens: students from culturally and linguistically different families and student with disabilities. Figures 2 and 3 share the breakdown of nominated students for the current 2020–2021 school year. Figure 2, "Javits Grants TOPS Nominations by Ethnicity/Race," shows that 39.9% of the students seen as having high potential were Black/African American, 36.6% were Hispanic, 11.0% were Asian, 8.7% were White, 3.5% were of mixed race, and 0.3% were American Indian or Alaska Native. As noted earlier, approximately 84% of students in Milwaukee Public Schools are students of color. Figure 2 shows that, of students recognized using the TOPS, 81% were Black/African American, Hispanic, Indigenous, and multi-race students. The pattern of nominations for children of color indicates that teachers are viewing historically underrepresented students through a strength-based lens; seeing potential in students who were formerly viewed as at risk.

As part of the U-STARS model, teachers offer challenging and enriched learning opportunities to all students, scaffolding these upward depending on each student's response and needs. The TOPS allows teachers to systematically collect observational data points on how their students are responding to these differentiated experiences and provides a document to support portfolio building to show evidence of student achievement (Coleman et al., 2010) [8]. Because the TOPS is used by teachers in their own classrooms, during lessons which are intentionally designed to be challenging and enriching, it gives teachers a chance to see firsthand what students are capable of accomplishing. The TOPS is not meant to be a stand-alone instrument; it is used in conjunction with other identification tools to help gain a fuller understanding of student's strengths.

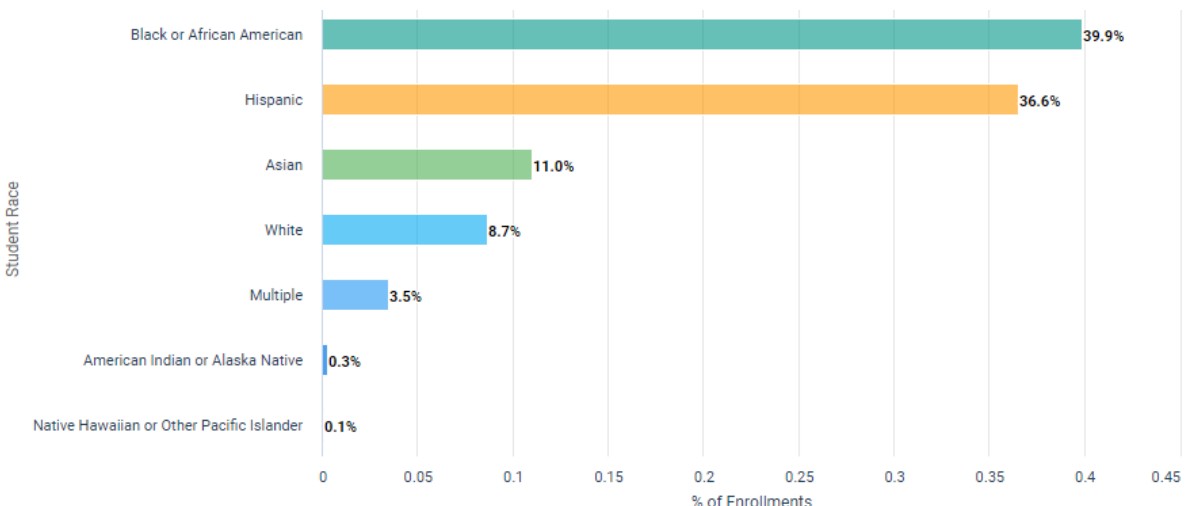

**Figure 2.** Javits Grants TOPS Nominations by Ethnicity/Race.

## Javits Grants TOPS Nominations by Twice Exceptionality

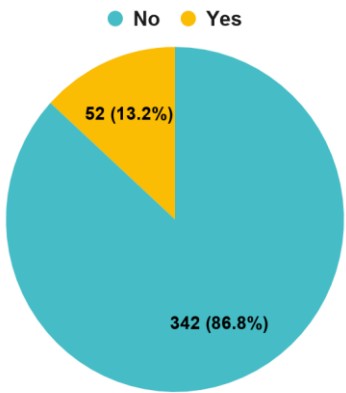

**Figure 3.** Javits Grants TOPS Nominations by Twice Exceptionality.

Figure 3, "Javits Grants TOPS Nominations by Twice Exceptionality," shares the 2020–2021 nomination pattern for teacher nominations of students who are twice exceptional (those who have both a disability and a clear area of strength). Again, we see that teachers' views of students with special education needs are beginning to include a strength-based lens as well with 13.2% of nominations being students who are twice exceptional.

As the numbers of students being recognized as having high potential increase, the district has also expanded opportunities to nurture and respond to the needs of these students. These opportunities include the enrichment and differentiated instruction provided within the classroom and school as well as opportunities provided at the district level. Dorothy Schuller, who started the first camps, states,

> These opportunities were available for students in other districts, and we recognized that our students deserved them too. We were able to create highly engaging camps for our students at no cost through the grants.

The Saturday and summer camps have given students with high potential a chance to come together and share their interests. This shared experience can be critical for bright students of color, many of whom may feel marginalized within their home schools (Gallagher et al., 2020) [11]. The table below (Table 1) shows the camps that were available to the students as well as the shift to virtual camps during the COVID-19 restrictions. The camps provided 658 enriched learning experiences for MPS students. (Note: Students likely participated in more than one camp; this is not an unduplicated count.) Each camp was hosted by a different SEE US! school, which allowed teachers to grow their professional

practices using STEM, creativity, inquiry, and questioning strategies. Susan O'Brien, MPS Javits grant coordinator, says,

> The camps served as a hands-on professional development opportunity. The same engaging practices that the teachers used with students with gifted potential at the Saturday camps became regular practices in the classrooms.

**Table 1.** Bridge to Greater Opportunities: Saturday and Summer Camps. Numbers of Student Participants at Each Camp.

| Bridge to Greater Opportunities: Saturday and Summer Camps. Numbers of Student Participants at Each Camp |
| --- |
| Summer 2016—First Summer Camp Imagination: Learn to Be an Inventor (24 students) |
| Fall 2016—Saturday Camp: STEM Challenges—Designing a Park (33 students) |
| Spring 2017—Saturday Camp: Peeps Design Challenges (45 students) |
| Summer 2017—Summer Camp: Fairy Tale STEM Challenges (54 students) |
| Fall 2017—Saturday Camp Create, three locations: Tall Tower Challenges (46 students) |
| Spring 2018—Saturday Camp Create, three locations: Design Challenges (39 students) |
| July 2018—Camp Imagination: Water-Themed Creativity Camp (22 students) |
| October 2018—Creativity Takes Courage, Design Challenge: Build a Better Candy Bag (32 students) |
| November 2018—Creativity Takes Courage (27 students) |
| March 2019—Camp Imagination, three sites: Build a Windmill/Wind Power (31 students) |
| April 2019—Creativity Takes Courage, Spring Theme (29 students) |
| June 2019—Summer Camp, two sites: Let's Go Camping (40 students) |
| November 2019—Creativity Takes Courage, STEM Saturday Camp, four sites: Fairy Tale STEM Challenges (51 students) |
| December 2019—Creativity Takes Courage, STEM Saturday Camp, four sites: STEM Architecture Challenges (64 students) |
| July 2020—Virtual STEM Camp, two weeks of camp: Technology and Coding Camp (9 students) |
| April 2021—Virtual Camp Imagination, Talent Development Camp, three sites: Science themes "Over the Rainbow" and "Planes, Trains, and Automobiles" (28 students) |
| Spring 2021—Three Student Leadership Academies (talent development for student leaders) (84 students) |
| Summer 2021—Summer Camp, two weeks of camp: Young Ornithologists and STEM Carnival |
| Number of Students Who Participated across the Camps = 658 * |

* Students may have participated in more than one camp.

## 14. School Impact

One of the greatest impacts of the SEE US! grant and emerging in the SURGE grant was the nomination of students with high ability/high potential and the "at potential" mindsets that emerged in classrooms and schools as a whole. Teachers adopted a strength-based approach to emergent talent in young learners. It was not uncommon for a teacher to say, "I never looked at him like that before," or "I thought he was just naughty". Teachers learned to see high ability/high potential in students with high rates of absenteeism, behavioral challenges, or special education labels.

Family engagement increased at each school with SEE US! implementation. The project aimed to support families of students in SEE US! demonstration classrooms. The support offered to families included the U-STARS~PLUS Family Science Packets, home libraries, and gifted and talented family events. Parents and guardians of TOPS-nominated students were invited to a Gifted 101 session yearly to learn about their child's areas of high ability/high potential as well as opportunities afforded to them as identified students. Teachers were able to attend state and national gifted and talented conferences. A select group of SEE US! participants was able to visit a local school featured in The Curious Classroom, a book we used for professional development. The monthly meetings allowed teachers to collaborate, speak about students with a positive mindset, and stay focused on grant goals.

## 15. Student Impact

SEE US! students have benefited from access to U-STARS~PLUS lessons, hands-on learning, and high-quality materials that otherwise might not have been affordable. This included each school getting libraries of all the U-STARS~PLUS trade books and the culturally responsive U-STARS~PLUS companion books. Due to professional development,

materials, and structured support, more hands-on science lessons took place, thus increasing student engagement. Students were also able to develop their curiosity by learning in classrooms where there were wonder walls, wonder journals, higher-level questioning techniques, and an inquiry approach to instruction. Most importantly, students enjoy classrooms where their assets and talents are recognized and developed, and what previously was seen as a deficit might now give insight to teachers and provide an entry point for student learning.

## 16. Current Definition of and Programming for Gifted and Talented (GT) Students in MPS

Milwaukee Public Schools recently revised its identification goals for gifted and talented (GT) programs. It states the following:

> Giftedness is fluid, inclusive, and is exhibited across gender, race, ethnicity, income level, languages, and exceptionality. Gifted students exist in all MPS schools. The composition of the students identified for gifted and talented services will reflect the total student population in the district. Identification is responsive and it is acknowledged that student needs may change over time to be exhibited in a particular context.

As a result, MPS adopted a multifaceted conception of giftedness and multidimensional identification practices. The MPS 2018 revised Gifted and Talented Identification Process policy states the following:

> Students can be identified for GT through five areas: academic, artistic, creativity, leadership, intelligence. MPS has a specific process and universal screeners for identifying students in academics and intelligence. Creativity, artistic, and leadership identification processes are being developed. Staff members look at multiple criteria that are appropriate for each category of giftedness before determining whether special GT services are needed. When students are screened for services, MPS is committed to identifying students that have been traditionally overlooked with unmet needs.

The current MPS Gifted and Talented Programming Guidelines use multiple criteria to screen for and identify giftedness:

1. The Cognitive Abilities Test (CogAT). The CogAT assesses students' verbal, quantitative, and nonverbal reasoning abilities. All second-grade students in MPS will be assessed through the CogAT during the first semester and all other students on an as-needed basis. Students who score in the ninth stanine in any domain will be flagged as GT and receive services as needed.
2. Teacher's Observation of Potential in Students (TOPS). This tool provides evidence of a student who learns easily, shows advanced skills, displays curiosity and creativity, has strong interests, shows advanced reasoning and problem solving, displays spatial ability, shows motivation, shows social perceptiveness, and displays leadership.
3. The Star 360 universal screener. All students will screen using Star Math and Reading assessments three times a year. Students are recommended for Tier 2 GT services if their performance is in the national 75th percentile or above in either math or reading. Student performance in the national 95th percentile or above or top 5th percentile using local school data in either math or reading is recommended for Tier 3 GT services. Students scoring in the top 75th percentile and above will be identified as significantly above benchmarks in the MPS data warehouse and will receive GT services as needed.
4. Evidence of demonstrated performance. Such evidence, which can be used in education planning, includes artifacts that consistently demonstrate that the student is working at advanced levels.
5. Classroom data. Evidence includes exceptional grades, projects, participation, performance, formative assessments, creativity, and leadership demonstration.

6. Parent input. This is a valued piece of information when reviewing student data. Parents are encouraged to complete the "Parent Input Form for Gifted and Talented Identification" so that teacher teams have parents' input into the education planning for their child.

Milwaukee Public Schools uses the Wisconsin Department of Public Instruction's Response to Intervention (RTI) programming model. RTI is a process for achieving higher levels of academic and behavioral success for all students through high-quality instruction practice, continuous review of student progress, and collaboration. We are currently developing implementation practices for this model according to tiers. GT students have a Tier 2 or Tier 3 plan documented, and progress is monitored.

1. Tier 1 services occur in the regular classroom with modifications and differentiation made by the classroom teacher. The classroom teacher provides these services.
2. Tier 2 services occur through curriculum compacting, flexible grouping, problem-based learning, and/or tiered lessons, which may change over time and are dependent upon the needs of the student. These services may be provided by the classroom teacher, academic coach, and/or other trained staff members.
3. Tier 3 services occur as an individualized program, which may or may not mean grade acceleration, subject acceleration, pull-out services, specialized programs, and curriculum compacting. Tier 3 activities require a parent conference with the classroom teacher, the school administrator, and/or another authorized staff member such as the school psychologist, counselor, or RTI specialist.

Supports and services for gifted and potentially gifted students in MPS have come a long way and are still evolving. We have learned a lot on our journey toward building excellence with equity across our gifted programs.

## 17. Lessons Learned on This Journey

As with any major undertaking, we feel that we have learned a lot. Systemic change takes commitment and passion. It requires trust building to ensure deep stakeholder ownership, and it entails systematic capacity building to develop the knowledge and skills needed to support the changes being made. The following are some of the lessons we have learned as we have moved along this journey:

1. Increasing access to opportunities as well as increasing the identification of gifted potential in underrepresented communities is possible. However, for this to occur there must be a change in mindset in educators, principals, and all stakeholders. This means rejecting the idea that children from minority communities, who often attend underserved schools, lack gifted potential. It requires implementing culturally relevant teaching practices that focus on developing the strengths that students possess rather than focusing on deficit models that result in being unable to recognize *true* gifted potential.
2. Using U-STARS~PLUS as a pedagogical framework to enhance teaching practices with a more hands-on approach has proved to be effective. Teachers who used the U-STARS~PLUS model manifested an increase in student engagement; a decrease in behavioral problems; and a greater satisfaction, joy, and enthusiasm in their teaching practices.
3. The success that MPS has had in increasing identification and in meeting the needs of gifted potential in students relies on being able to capitalize on what is working rather than starting from scratch. This includes using teachers' strengths, developing a cohesive approach to the implementation of the grants, the participation of diverse stakeholders, the adequate use of financial assets, consulting with experts in the field of gifted education, and—first and foremost—the buy-in from schools and parents.
4. Teachers and school principals play a key role in the implementation of innovative strategies that seek to identify, nurture, and develop gifted potential in students from underrepresented communities.

5. Program design that includes a position specific to these grants is essential for implementation with fidelity.

6. The implementation of the Expanding Excellence, SEE US!, and SURGE grants has increased access to services for all students at schools, serving as a bridge to greater opportunities to learn.

7. Even though much progress needs to be made at the district level to have a more sequenced and consistent process of identification, huge gains have been made. This includes adopting a more multifaceted conception of giftedness and multidimensional identification practices at the district level, drastically increasing the identification of students with gifted potential, and developing teacher capacity and abilities to identify students' pleasing and non-teaching pleasing behaviors as "at potential" rather than as "at risk".

8. The inclusion of parents as active participants in the nomination process and nurturing of students' gifted potential has contributed to establishing stronger relationships between parents, teachers, and district leaders.

9. Using TOPS as a non-normative tool to identify gifted potential in minority students has been effective. This was reflected in the increase of students' nominations (twice as many as with the CogAT alone) as well as a change in teachers' perspectives about giftedness. TOPS helped teachers see their students through a strength-based lens.

## 18. Conclusions

Shifting the education system toward a strength-based approach that supports the expansion of excellence to *all* students and ensuring that each student receives an appropriately stimulating and enriched education is a major undertaking. MPS maintains a long-term commitment to identifying and serving all students equitably using the RTI framework with particular interest in underrepresented populations and closing the excellence gap. While we have made substantial progress, we know that our journey must continue.

**Author Contributions:** Conceptualization: S.O., M.R.C., D.L.S., M.A.L. and G.D. contributed to the conceptualization of the article and the original draft preparation. S.O. lead on writing reviews and editing, project administration, funding acquisition. All authors have read and agreed to the published version of the manuscript.

**Funding:** This program was funded by the Javits Gifted and Talented Students Education Grant Program-United States Department of Education.

**Institutional Review Board Statement:** This manuscript was developed as part of a standard program review and therefore an IRB was not required. District approval was secured for publication and participants consented to the use of their input within the manuscript. Please let us know if any further information is needed.

**Informed Consent Statement:** Informed consent was obtained from all subjects in a variety of manners including surveys, interviews, and verbal consent.

**Data Availability Statement:** Evaluation data reported internally for formative assessment of program implementation. Data is not publicly reported.

**Conflicts of Interest:** The authors declare no conflict of interest.

# Appendix A

**Table A1.** Professional Development Offered to Grant Participants: Capacity Building for Change 2018–2021.

| Date | Title | Audience and Content |
|---|---|---|
| February 2018 | School Principals SEE US! Grant Information Session | School principals, regional superintendents<br>School principals SEE US! grant information |
| April 2018 | Grant Kick-Off | SEE US! demonstration classroom teachers, support staff, and administrators<br>Learning about the SEE US! grant and deepening our understanding of giftedness |
| 13–15 June 2018 | SEE US! Summer Institute: U-STARS~PLUS and TOPS | SEE US! demonstration classroom teachers, support staff, and administrators<br>U-STARS~PLUS, family engagement, and TOPS; best practices in gifted |
| 25 September 2018 | SEE US! Fall Workshop | SEE US! demonstration classroom teachers, support staff, and administrators<br>TOPS, nurturing creativity, instruction strategies |
| 28 March 2019 | SEE US! Workshop | SEE US! demonstration classroom teachers, support staff, and administrators<br>Responding to the needs of high-ability/high-potential students in our classrooms |
| 2 May 2019 | SEE US! Workshop | SEE US! demonstration classroom teachers, support staff, and administrators<br>U-STARS~PLUS and culturally responsive practices; culturally responsive teaching practices; cluster grouping |
| 18 June 2019 | SEE US! Summer Institute: Day 1 | SEE US! demonstration classroom teachers, support staff, and administrators<br>Inquiry-based instruction to cultivate a curious classroom |
| 19 June 2019 | SEE US! Summer Institute: Day 2 | SEE US! demonstration classroom teachers, support staff, and administrators<br>Differentiated hands-on learning:<br>Grades 1 and 2—mini-inquiries<br>Grade 3—document-based questions (DBQs) |
| 14 September 2019 | New SEE US! Teacher Grant Kick-Off Event | New SEE US! demonstration classroom teachers, support staff, and administrators<br>Overview of grant goals; U-STARS~PLUS and TOPS; panel of experienced SEE US! teachers with time to collaborate |
| 10 October 2019 | SEE US! Fall Workshop | SEE US! demonstration classroom teachers, support staff, and administrators<br>Recognizing students with potential, creating observable environments, setting the stage for inquiry |
| 11, 12 December 2019 | SEE US! Inquiry Workshop | SEE US! demonstration classroom teachers, support staff, and administrators<br>Bringing inquiry to life in your classroom |
| Spring 2020 | SEE US! Spring Workshop (cancelled due to COVID-19) | |
| 22 June 2020 | SEE US! Virtual Summer Workshop: Day 1 | SEE US! demonstration classroom teachers, support staff, and administrators<br>Designing inquiry experiences; reviewing U-STARS~PLUS hands-on science, questioning strategies, and family engagement; social studies inquiry |

**Table A1.** *Cont.*

| Date | Title | Audience and Content |
|---|---|---|
| 23 June 2020 | SEE US! Virtual Summer Workshop: Day 2 | SEE US! demonstration classroom teachers, support staff, and administrators<br>Working with school teams to create inquiry lessons and units |
| 24 June 2020 | SURGE Virtual Workshop: Day 1 | SURGE demonstration classroom teachers, support staff, and administrators<br>TOPS training |
| 25 June 2020 | SURGE Virtual Workshop: Day 2 | SURGE demonstration classroom teachers, support staff, and administrators<br>Using STEM instruction to create an observable environment |
| 26 June 2020 | SURGE Virtual Workshop: Day 3 | SURGE demonstration classroom teachers, support staff, and administrators<br>Meeting student needs and the RTI framework, grade-level planning, and ordering of classroom |
| 26 September 2020 | SEE US! and SURGE Professional Development: Using TOPS in a Virtual Environment | SEE US! and SURGE demonstration classroom teachers, support staff, and administrators<br>Saturday—Added session due to COVID-19 virtual teaching: virtual environments for TOPS observations |
| 28, 29 September 2020 | SEE US! and SURGE: Creating an Observable Environment to Recognize High- Ability/High-Potential Students | SEE US! and SURGE demonstration classroom teachers, support staff, and administrators<br>After-school sessions—Added due to COVID-19 virtual teaching: virtual environments for TOPS observations |
| 22 October 2020 | SURGE Fall Workshop | SURGE demonstration classroom teachers, support staff, and administrators<br>Using TOPS with confidence; nurturing, recognizing, and responding during difficult times; recognizing potential in virtual settings; using STEM in virtual settings |
| 28 October 2020 | SEE US! Fall Workshop | SEE US! demonstration classroom teachers, support staff, and administrators<br>Using TOPS with confidence; nurturing, recognizing, and responding during difficult times; recognizing potential in virtual settings; using inquiry in virtual settings |
| Spring 2021 | SEE US! and SURGE Spring Workshop (delayed to summer due to COVID-19 meeting restrictions) | |
| 21–23 June 2021 | SEE US! Summer Workshop | SEE US! demonstration classroom teachers, support staff, and administrators<br>Inquiry workshop: Heidi Mills; bringing curriculum to life through inquiry |
| 24, 25 June 2021 | SURGE Cohort A Summer Workshop | SURGE demonstration classroom teachers, support staff, and administrators<br>Best practices in clustering for student readiness |
| 28–30 June 2021 | SURGE Cohort B Summer Workshop | SURGE demonstration classroom teachers, support staff, and administrators<br>Introduction to grant, identifying gifted potential, and providing services to students |

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
