# Peer review of "Milwaukee Makes a Difference: Recognizing Gifted Students from Culturally and Linguistically Diverse Families"

_education, doi:10.3390/educsci11100578_

Round 1

Reviewer 1 Report

This paper presents an interesting approach implemented in the U.S. to better identify and nurture talents and gifts in underprivileged students. This is a case report and not an empirical study, so I only have some questions and suggestions for minor improvement.

Abstract:

It might be informative for international readers to clarify that “across all 50 states” means the United States.

Shouldn’t it read “Milwaukee Public Schools” (plural)?

Main text:

It might be helpful for readers who are not Americans to explain what MPS actually is. Is it a program? Is it an institution?

The same is true for TOPS. After reading through the article, I did not have the feeling that I know what TOPS actually is. I see that there is a reference to another source that probably explains what it is. However, it would be good to explain it in more detail also in this article, at least in a few sentences.

ll. 140-141: What is the difference between gifted students, talented students, and students with high potential?

In l. 145 the font suddenly changes.

Fig. 2 reports the shares of students with different ethnicities who were nominated within TOPS. Those numbers are taken as evidence for the positive effects of TOPS. However, wouldn’t it be more informative to set those numbers in relation to the base rates of ethnicities in those schools? I was also wondering whether it was checked how valid the nomination results were. Was there a criterion that could be used for a validity check?

If possible, it might be interesting to include some experience reports from students who participated in the program.

What is the difference between talent/gift in academic areas and intelligence? Cognitive ability is a powerful predictor of academic achievement.

I was wondering how “[creativity], artistic, and leadership identification processes are being developed” (ll. 587-588) or how their validity will be tested. Measuring creativity, artistic talents, and leadership talent is a very difficult task. Related to this point, I was wondering how exactly TOPS “provides evidence of a student who learns easily, shows advanced skills” etc. (ll. 599-603) and how TOPS contributes to the identification process above and beyond cognitive ability tests (as far as the identification of cognitive/academic talents/gifts is concerned).

ll. 685-686: As TOPS seems to address much more than cognitive abilities, isn’t it clear that it will result in more nominations than the CogAT alone? And here again, I wondered whether nominations via TOPS are also more valid than nominations via the CogAT. Higher nomination rates might not be a value in itself if the nominations are invalid.   

Author Response

Thank you for your thoughtful review of this manuscript. Please find our responses below:

  1. we have clarified that this description is in the United States and that Milwaukee Public Schools is a large urban district in Wisconsin
  2. We have added some information about the TOPS to help explain what it is, however a full description of this and U-STARS is beyond the scope of this paper.
  3. The added the numbers of students of color for the MPS district as a comparison point for the students nominated with the TOPS. Given that this was the "nurturing phase" of support for students the process was intentionally inclusive.
  4. While it would be great to have some direct reports from students, and this would be a wonderful follow-up - it was beyond the scope of this paper.
  5. We also regret  that an in-depth discussion of the differences between talent/giftedness/ and intelligence is also not possible within the paper.
  6. It is difficult to measure and validate creativity, artistic and leadership as areas of talent/giftedness. These areas often go well beyond the standard measures of academic ability to include portfolios, adjudications of work, and documented accomplishments. The TOPS does include observations behaviors that indicate a child learns easily (e.g. shows strong memory, carries out complex instructions with ease) and advanced skills (e.g. reads and comprehends on an advanced level (this may be seen in listening comprehension), communicates well with symbols (art, design, music, or dance)). 
  7. Yes, the TOPS does address more then cognitive abilities and yes this makes it particularly useful to move beyond the CogAT to explore other areas of strenght that we may see in our students. Again, this is to establish a pool of students for nurturing students potential.
  8. Higher nomination rates open the opportunity for further enrichment and the chance to document strenghts of the student - they do not necessarily mean a student will be formally identified.

Reviewer 2 Report

The manuscript is focused on the problematic of gifted students. This is very actual problematic, not only at universities among researchers abut also among teachers, parents and also all society. Authors introduced very successful text, where is possible to find about effort, how to have success in this field. I have got only three comments.

  1. Please do not use pie charts, it is not appropriate for scientific journals.

  1. Please rewrite text or eliminate some kinds of information, mainly in the part, where the presentation of figures is. The kinds of information are also in the text and also in the figure, so they are duplicated.

  1. Please use passive in whole text.

I hope my comments are helpful.

Author Response

Thank you for taking time to review and offer feedback on this manuscript. 

  1. While a pie chart may be unusual for this journal it is the easiest visual way to show the data and so we feel it serves the purpose well.
  2. We have modified some of the text around the figures - unfortunately the second review requested more detailed information and so the text has been expanded not shortened.
  3. We have used active voice to enhance the narrative as a description of the project is important.